Photobacterium sanctipauli sp. nov. isolated from bleached Madracis decactis (Scleractinia) in the St Peter & St Paul Archipelago, Mid-Atlantic Ridge, Brazil

Moreira Ana Paula B. 1
Duytschaever Gwen 1
Chimetto Tonon Luciane A. 1
Fróes Adriana M. 1
de Oliveira Louisi S. 1
Amado-Filho Gilberto M. 2
Francini-Filho Ronaldo B. 3
De Vos Paul 4 5
Swings Jean 4 5
Thompson Cristiane C. 1
Thompson Fabiano L. 1 6 fabiano.thompson@biologia.ufrj.br
1 Institute of Biology, Federal University of Rio de Janeiro (UFRJ) , Rio de Janeiro , Brazil
2 Botanical Garden Research Institute (JBRJ) , Rio de Janeiro , Brazil
3 Department of Environment and Engineering, Federal University of Paraíba (UFPB) , Brazil
4 BCCM/LMG Bacteria Collection, Ghent University , Ghent , Belgium
5 Laboratory of Microbiology, Faculty of Sciences, Ghent University , Ghent , Belgium
6 Laboratório de Sistemas Avançados de Gestão de Produção - SAGE - COPPE, Federal University of Rio de Janeiro , Rio de Janeiro , Brazil
Weightman Andrew
Electronic publication date: 2014 Jun 19
Publication date: 2014
Volume: 2
Electronic Location ID: e427
Received 2014 Mar 18; Accepted 2014 May 22
Copyright: © 2014 Moreira et al.
Copyright year: 2014
Copyright holder: Moreira et al.
License: This is an open access article distributed under the terms of the Creative Commons Attribution License, which permits unrestricted use, distribution, reproduction and adaptation in any medium and for any purpose provided that it is properly attributed. For attribution, the original author(s), title, publication source (PeerJ) and either DOI or URL of the article must be cited.
License URL: https://creativecommons.org/licenses/by/4.0/

Keywords: Photobacterium sanctipauli, St Paul’s rocks, Coral bleaching, New species, Genomic taxonomy

Funding: CNPq FAPERJ Federal Public Service for Science Policy, Belgium This work was supported by CNPq grants to APBM, GD, GMAF, RBFF and FLT. LACT received a grant from FAPERJ. The BCCM/LMG Bacteria Collection is supported by the Federal Public Service for Science Policy, Belgium. The funders had no role in study design, data collection and analysis, decision to publish, or preparation of the manuscript.

==============================
Five novel strains of Photobacterium (A-394T, A-373, A-379, A-397 and A-398) were isolated from bleached coral Madracis decactis (scleractinian) in the remote St Peter & St Archipelago (SPSPA), Mid-Atlantic Ridge, Brazil. Healthy M. decactis specimens were also surveyed, but no strains were related to them. The novel isolates formed a distinct lineage based on the 16S rRNA, recA, and rpoA gene sequences analysis. Their closest phylogenetic neighbours were Photobacterium rosenbergii, P. gaetbulicola, and P. lutimaris, sharing 96.6 to 95.8% 16S rRNA gene sequence similarity. The novel species can be differentiated from the closest neighbours by several phenotypic and chemotaxonomic markers. It grows at pH 11, produces tryptophane deaminase, presents the fatty acid C18:0, but lacks C16:0 iso. The whole cell protein profile, based in MALDI-TOF MS, distinguished the strains of the novel species among each other and from the closest neighbors. In addition, we are releasing the whole genome sequence of the type strain. The name Photobacterium sanctipauli sp. nov. is proposed for this taxon. The G + C content of the type strain A-394T (= LMG27910T = CAIM1892T) is 48.2 mol%.

Introduction

Currently the genus Photobacterium comprises 26 formally described species (Euzéby, 2013; Liu et al., 2014; Srinivas et al., 2013). The habitats and isolation source include seawater (Reichelt, Baumann & Baumann, 1976; Yoshizawa et al., 2009), sea sediments (Jung et al., 2007; Seo et al., 2005a; Yoon et al., 2005), saline lake water (Rivas et al., 2006), and a variety of marine organisms with which the strains associate as commensals, saprophytes, bioluminescent symbionts, or pathogens (Urbanczyk, Ast & Dunlap, 2011). The list of hosts include fish (Liu et al., 2014; Onarheim et al., 1994; Ruimy et al., 1994), oyster and crab (Gomez-Gil et al., 2011), amphipods (Bartlett & Welch, 1995), sea hare (Seo et al., 2005b), squid (Kaeding et al., 2007) zoanthids (Palythoa caribaeorum) (Chimetto et al., 2010) and corals. P. jeanii and P. rosenbergii were the previously described species isolated from corals (Chimetto et al., 2010; Thompson et al., 2005b). P. jeanii was associated with healthy colonies of the scleractinian Merulina ampliata in Australia and the octocoral Phyllogorgia dilatata in Brazil (Chimetto et al., 2010), whereas P. rosenbergii was retrieved from several scleractinians, including healthy Pachyseris speciosa and diseased M. ampliata, P. speciosa and Barabattoia amicorum, in Australia (Thompson et al., 2005b), as well as from healthy Mussismilia hispida in Brazil (Chimetto et al., 2009). Photobacterium strains found in association with corals (healthy Acropora palmata) were identified as P. phosphoreum, P. damselae and P. mandapamensis (Ritchie, 2006). Coral microbiologists are challenged to increase our understanding in order to mitigate the worldwide spread of infectious diseases that are implicated in the decrease of coral cover in reef systems, markedly associated with climate changes and anthropogenic driven environmental disturbances (De’ath et al., 2012; Eakin et al., 2010; Mouchka, Hewson & Harvell, 2010; Rosenberg et al., 2007).

The study of the culturable heterotrophic microbiota of healthy and bleached Madracis decactis in the Brazilian St Peter & St Paul Archipelago (SPSPA) analyzed 403 isolates (Moreira et al., 2014). P. angustum and P. damselae were retrieved from healthy colonies, whilst five novel Photobacterium strains were only retrieved from the bleached corals. These five novel isolates originated from two colonies, but shared nearly identical 16S rRNA gene sequences. They showed less than 97% 16S rRNA gene sequence similarity towards the closest phylogenetic neighbor, Photobacterium rosenbergii (Moreira et al., 2014).

The present study aimed to describe a novel Photobacterium species, represented by five strains previously isolated in the SPSPA (Table S2), based on a polyphasic approach.

Materials and Methods

All strains were isolated using thiosulfate-citrate-bile salt-sucrose (TCBS) medium at ambient temperature (∼27 °C) after 24–48 h incubation (Moreira et al., 2014). Gene sequences of 16S rRNA, recombination repair protein (recA), and RNA polymerase alpha subunit (rpoA) were obtained as described previously (Chimetto et al., 2009; Chimetto et al., 2008; Moreira et al., 2014). Primers used for gene amplification and sequencing were described in Moreira et al. (2014), Sawabe, Kita-Tsukamoto & Thompson (2007), Thompson et al. (2005a) and Thompson et al. (2001). Raw sequence data were transferred to ChromasPro V. 1.7.1 (Technelysium Pty. Ltd, Tewantin, Australia) where consensus sequences were determined. Pairwise similarities of these sequences with those of the closest phylogenetic neighbours were calculated using Jalview V.2 (Waterhouse et al., 2009). Sequences were aligned using ClustalW. Similarity matrices and phylogenetic analysis were performed by using the MEGA (Molecular Evolutionary Genetics Analysis) version 5.2 software (Tamura et al., 2011). Trees were drawn using the neighbour-joining method (Saitou & Nei, 1987). The robustness of each topology was checked by 1,000 bootstrap replications (Felsenstein, 1985). For genome sequencing 1ng of high quality DNA (obtained as in Moreira et al. (2014)) was used to construct the genomic paired-end library using the Nextera XT Sample Preparation Kit (Illumina®). Through this method, the DNA was simultaneously fragmented and tagged with sequencing adapters. The library size distribution was accessed using the 2100 Bioanalyzer and the High Sensitivity DNA Kit (Agilent®). The accurate quantification of the library was accomplished using the 7500 Real Time PCR (Applied Biosystems®) and the KAPA Library Quantification Kit (Kapabiosystems®). Paired-end (2 ×250 bp) sequencing was performed on a MiSeq (Illumina®) using the MiSeq reagent kit v2 (500 cycles). R1 and R2 reads were quality filtered (Q > 20) and 3’ end trimmed with Prinseq v0.20.4 (Schmieder & Edwards, 2011). Ray v. 2.3.1 was used to perform De novo assembly into scaffolds and contigs with default parameters (Boisvert et al., 2012). General genome features were determined through Rapid Annotations Using Subsystems Technology (The RAST server version 4.0) (Aziz et al., 2008). In silico DDH values were estimated to one strain of each Photobacterium species with publicly available genome using GGDC 2.0 (Auch, Klenk & Göker, 2010; Auch et al., 2010). This online tool infers genome-to-genome distances between pairs of entirely or partially sequenced genomes. Intergenomic distances are employed for wet-lab DDH prediction. Briefly, genome pairs were aligned with BLAST+ (Camacho et al., 2009) to generate a set of high-scoring segment pairs (HSPs). The information they contained (e.g., the total number of identical base pairs) was transformed into a distance value by the best-fit formula, according to (Meier-Kolthoff et al., 2013). DDH prediction from intergenomic distance, including confidence intervals, were provided by a tested generalized linear model (GLM, Nelder & Wedderburn, 1972) with log transformation (Meier-Kolthoff et al., 2013). Strains and genome accession numbers are in Table S1. AAI was calculated (according to Konstantinidis & Tiedje (2005)) towards the closest neighbor species determined by RAST (P. leiognathi). The gene sequence data obtained in this study are available through the open access website TAXVIBRIO (http://www.taxvibrio.lncc.br/). The GenBank accession numbers for the 16S rRNA, recA, and rpoA genes and genome sequences are listed in Table S1. The mol% G + C was determined according to Moreira, Pereira & Thompson (2011). MALDI-TOF MS protein profiles were determined as described previously (Wieme et al., 2012). Isolates were subcultured twice on MA for 24h at 30 °C. MALDI-TOF MS was conducted using a 4800 Plus MALDI-TOF/TOFTM Analyzer (Ab Sciex NV) in linear mode and the 4000 Series Explorer Software v3.5.3 (Applied Biosystems®). Spectra were generated with mMass software v5.5.0 (Strohalm et al., 2010). Type strains of the three closest related Photobacterium species were included for comparison. Phenotypic characterization was performed using commercial miniaturized kits (API 20E, API NE and API ZYM; BioMerieux) as described previously (Chimetto et al., 2010; Kim et al., 2010; Thompson et al., 2005b) and by BIOLOG GEN III metabolic fingerprinting (Biolog), following the manufacturer’s instructions. These tests included determination of temperature, pH and salinity growth ranges, several biochemical responses and 71 carbon source utilization assays. Unless indicated otherwise, isolates were grown onto MA for 24 hr at 30 °C. The optimal growth temperature was determined using TSB supplemented with 2.0% NaCl at pH 7.5, the optimal pH was determined in TSB supplemented with 2.0% NaCl at 30 °C and the optimal salinity was determined in peptone water (1.5% Peptone, 30 °C, pH 7.5). Growth under anaerobic conditions was determined after incubation in an anaerobic atmosphere (Microanaerobac, PROBAC, Brasil) on MA at 30 °C. Fatty acid methyl ester analyses were performed using the Sherlock Microbial Identification System (Royal Life Sciences Pvt. Ltd) according to the standard protocol. To this end, isolates were harvested from MA after 24 h of incubation at 30 °C. The results of these phenotypic analyses are presented in the species description and the distinctive features in Table 1.

Table 1 Phenotypic differences between P. sanctipauli sp. nov. and related Photobacterium species.

Taxa: 1, P. sanctipauli sp. nov. (five strains); 2, P. rosenbergii LMG 22223T (Srinivas et al., 2013; Thompson et al., 2005b); 3, P. gaetbulicola Gung 47T (Kim et al., 2010); 4, P. lutimaris LMG 25278T (Chimetto et al., 2010; Jung et al., 2007); 5, P. jeanii LMG 25436T (Chimetto et al., 2010; Srinivas et al., 2013); 6, P. leiognathi LMG 4228T (Baumann & Baumann, 1984; Chimetto et al., 2010; Nogi, Masui & Kato, 1998; Yoshizawa et al., 2009). +, Positive; −, negative; w, weak; v, variable; nd, no data available. All taxa are negative for Gram stain, lysine- and ornithine- decarboxylase, L-arabinose and D-sorbitol utilization; and positive for oxidase and alkaline phosphatase. Data in parentheses are for the type strains.

Characteristic	1	2	3	4	5	6	
Salinity growth range (%)	1–8	1–7	0–8	1–6	0.5–4	0.5–6	
Optimum NaCl concentration (%,w/v)	2–3	2–6	2–5	2–3	0.5–2	nd	
Temperature growth range (°C)	15–42	15–35	10–40	4–41	15–37	nd-37	
Optimum temperature (°C)	30	20–30	30	25–30	28	(26)	
pH growth range	6–11	6–10	5–9	5–9	5–9	nd	
Optimum pH	7.5	7–8.5	7–8	7.5–8.5	7–8	nd	
Enzyme activity							
Catalase	w	(+)	+	w	+	(−)	
Esterase (C4)	v(−)	+	+	+	+	+	
Esterase lipase (C8)	v(−)	+	+	+	+	(+)	
Lipase (C14)	−	(+ )	+	−	+	−	
Leucine arylamidase	+	−	−	+	+	nd	
Valine arylamidase	−	+	−	−	w	−	
Cystine arylamidase	−	−	−	+	−	nd	
Trypsin	−	−	−	−	+	(w)	
Acid phosphatase	−	+	−	+	+	nd	
Naphthol-AS-BI phosphohydrolase	+	+	−	+	+	+	
α-galactosidase	−	(+)	−	−	−	−	
α-glucosidase	−	(+)	−	−	+	(−)	
N-acetyl-β-glucosaminidase	+	+	−	−	(−)	nd	
β-galactosidase	+	+	−	−	+	+	
Arginine dihydrolase	+	+	−	+	+	+	
Tryptophane deaminase	v(w)	−	−	−	−	(−)	
Indole production from tryptophan	v(−)	−	nd	+	−	(−)	
Acetoin production from sodium pyruvate	−	−	nd	(−)	(w)	+	
Gelatinase	−	−	nd	−	+	−	
Fermentation							
Amygdalin	−	+	nd	(+)	−	(−)	
Glucose	+	+	+	−	+	+	
Utilization as sole carbon source							
Citrate	v(−)	+	+	+	−	−	
D-Maltose	v(−)	(+)*	+	+	−	+	
D-trehalose	v(−)	(+)*	+	+	−	−	
D-Cellobiose	v(w)	(+)*	+	+	−	−	
Sucrose	v(−)	(+ )*	+	+	v(+)	−	
D-Raffinose	−	(−)*	+	+	nd	−	
D-Melibiose	v(−)	(+)*	+	(−)	+	−	
β-Methyl-D-Glucoside	v(−)	(+)*	nd	nd	nd	nd	
D-Mannose	+	(+)*	+	+	−	+	
D-Salicin	+	(+)*	−	+	nd	nd	
D-Fructose	v(w)	(+)*	−	+	nd	−	
L-Rhamnose	v(−)	(+ )*	−	−	−	−	
D-Mannitol	v(−)	(+)*	+	−	−	−	
Myo-Inositol	v(−)	(+)*	+	+	−	−	
Tween 40	−	(w)*	+	+	nd	−	
DNA G + C content (mol%)	48.2	47.6–47.9	50.6	48.3	49.8	41.6	
Fatty acids							
C16:0 iso	−	1.9	0.4	−	1.9–3.5	−	
C18:0	0.5–0.7	−	−	−	−	−	
Notes.

* Data from this study.

Results and Discussion

16S rRNA gene sequence analysis revealed that the five isolates formed a tight monophyletic branch affiliated to the genus Photobacterium (Fig. 1). The five novel isolates shared more than 99% 16S rRNA gene sequence similarity. The sequence similarities towards the closest neighbours (based on 16S rRNA) were below the threshold (97%) established for species definition (Stackebrandt & Goebel, 1994; Vandamme et al., 1996). P. rosenbergii and P. gaetbulicola showed 96.6% sequence similarity, whereas P. lutimaris showed 95.8%. Other closely related neighbours have not been validly described yet. These are the cases of P. atrarenae (Kim et al., 2011) and P. marinum (Srinivas et al., 2013). The phylogenetic analysis based on 16S rRNA, recA, and rpoA gene sequences (3,135 nt in total) confirmed that the isolates formed a distinct lineage related to P. rosenbergii and P. gaetbulicola (Fig. 2). The novel isolates shared less than 87.2%, 96.5%, and 94.1% similarity based on recA, rpoA, and concatenated gene sequences (16S rRNA, recA, and rpoA) with their closest neighbours, respectively. These levels of similarity are below the cut-offs determined to define a species of the family Vibrionaceae (Thompson et al., 2009; Thompson et al., 2005a). The similarity levels between the novel isolates (A-394T, A-373, A-379, A-397 and A-398) ranged from 99.8% to 100% based on recA. Their rpoA sequences were identical. Trees based on partial sequences of the housekeeping genes recA (855 bp) and rpoA (969 bp) also confirmed their phylogenetic position in the genus Photobacterium and revealed they constituted a separate branch, clearly indicating that they belong to a new Photobacterium species (Figs. S1–S2). General features of A-394T genome are supplied in Table S3. In silico DDH (%) values between A-394T and P. angustum S14, P. damselae subsp. damselae CIP 102761, P. halotolerans DSM18316, P. leiognathi lrivu.4.1 and P. profundum 3TCK were 21.5 (±2.34), 22.7 (±2.37), 20.3 (±2.31), 21.6 (±2.35) and 20.6 (±2.31) respectively. AAI between A-394T and P. leiognathi lrivu.4.1 CIP 102761 was 75%.

Figure 1 16S phylogenetic tree.

Neighbour-joining phylogenetic tree of Photobacterium species based on 16S rRNA gene sequences (1,525 nt) showing the position of P. sanctipauli sp. nov. The optimal tree with the sum of branch length = 0.35538897 is shown. The evolutionary distances were computed using the Jukes-Cantor method. All positions containing alignment gaps and missing data were eliminated only in pairwise sequence comparisons (Pairwise deletion option). Phylogenetic analyses were conducted in MEGA5. Bootstrap values (>50%) based on 1,000 resamplings are shown. Salinivibrio was used as outgroup. Bar, 1% estimated sequence divergence.

Figure 2 Neighbour-joining phylogenetic tree based on concatenated 16S rRNA, recA and rpoA gene sequences (3,135 nt) showing the position of P. sanctipauli sp. nov. The evolutionary distances were computed using the number of differences method and are in the units of the number of base differences per sequence. All positions containing alignment gaps and missing data were eliminated only in pairwise sequence comparisons (Pairwise deletion option). Phylogenetic analyses were conducted in MEGA5. Bootstrap values (>50%) based on 1,000 resamplings are shown. Vibrio maritimus R-40493T was used as outgroup. Bar estimate sequence divergence.

Several phenotypic features can be used to differentiate the novel species from its closest phylogenetic neighbors. The growth at pH 11, tryptophane deaminase activity, presence of the fatty acid C18:0, and absence of C16:0 iso (Table S4). MALDI-TOF MS protein profiles distinguished the novel strains among each other and from P. rosenbergii (LMG 22223T), P. gaetbulicola (LMG 27839T) and P. lutimaris (LMG 25278T) (Fig. S3). MLSA was more discriminative than MALDI-TOF and FAME for strain differentiation. Phenotypic and chemotaxonomic variation observed among the strains of the novel species indicate they are not clonal (Table S5 and Fig. S3). Based on the polyphasic analysis including MLSA, MALDI-TOF MS fingerprint profiles, chemotaxonomic and phenotypic tests presented in this study, we propose to classify the five isolates as a new species, Photobacterium sanctipauli sp. nov.

Description of Photobacterium sanctipauli sp. nov.

Photobacterium sanctipauli (sanctí pauli N.L. gen. n. sanctipauli of Saint Paul, after the St Peter & St Paul Archipelago).

Colonies are small, beige, irregular shaped, with smooth and translucent edge and 1–2 mm in diameter after 24 h at 28 °C on MA under aerobic conditions. On TCBS colonies are green, round with a smooth border and 2–3 mm in diameter. Cells are small bacilli measuring 2–3 µm in diameter, Gram-negative, motile, facultative anaerobic, oxidase and catalase-positive. Grows well between 20 and 30 °C but not at 4 and 45 °C. No growth occurs in the absence of NaCl, but grows well under NaCl concentrations of 1%–8% (w/v). Grows at pH 6-11. Positive for alkaline phosphatase, leucine arylamidase, naphtol-AS-BI-phosphohydrolase, N-acetyl-β-glucosaminidase, β-galactosidase and arginine dihydrolase; but negative for lipase (C14), valine arylamidase, cystine arylamidase, trypsin, α-chemotrypsin, acid phosphatase, α-galactosidase, β-glucuronidase, α-glucosidase, β-glucosidase, α-mannosidase, α-fucosidase, lysine decarboxylase, ornithine decarboxylase, H2S production, urease activity, acetoin production (Voges–Proskauer) and gelatinase. Variable reactions were obtained for esterase (C4) (−), esterase lipase (C8) (−), tryptophane deaminase (w) and indole production (−) (whenever variable within species, result for the type strain is in parentheses). Reduces nitrate to nitrite but not to N2. Positive for fermentation/oxidation of glucose and mannitol but negative for inositol, sorbitol, rhamnose, saccharose, amygdalin and arabinose. Melibiose (+) gave variable reactions. D-Salicin, α-D-glucose, D-mannose, D-galactose are used as sole energy sources. Does not utilize dextrin, D-raffinose, glycerol, N-acetyl-D-galactosamine, D-glucose-6-PO4, D-aspartic acid, D-serine, gelatin, glycyl-L-proline, L-alanine, L-arginine, L-aspartic acid, L-glutamic acid, L-pyroglutamic acid, L-serine, pectin, L-galactonic acid lactone, mucic acid, quinic acid, D-saccharic acid, p-hydroxy-phenylacetic acid, methyl pyruvate, D-lactic acid methyl ester, citric acid, D-malic acid, bromo-succinic acid, γ-amino-butyric acid, α-hydroxy-butyric acid, β-hydroxy-D,L-butyric acid, propionic acid, acetic acid and formic acid. The following reactions are variable within the species: citrate (−), D-maltose (−), D-trehalose (−), D-cellobiose (w), gentiobiose (−), sucrose (−), D-turanose (−), stachyose (−), α-D-lactose (−), D-melibiose (−), β-methyl-D-glucoside (−), N-acetyl-D-glucosamine (−), N-acetyl-β-mannosamine (−), N-acetyl neuraminic acid (−), D-fructose (−), 3-methyl glucose (w), D-fucose (w), L-fucose (w), L-rhamnose (−), inosine (−), D-sorbitol (−), D-mannitol (−), D-arabitol (−), myo-inositol (−), D-glucose-6-PO4 (−), L-histidine (w), D-galacturonic acid (−), D-gluconic acid (−), D-glucuronic acid (−), glucuronamide (w), L-lactic acid (−), α-keto-glutaric acid (w), L-malic acid (−), tween 40 (−) and acetoacetic acid (w). Does not assimilate any of the substrates included in the API 20 NE system. The most abundant cellular fatty acids are summed feature 3 (43.5%; comprising C16:1ω7c and/or iso-C15 2-OH), C16:0 (21.4%), C18:1ω7c (11.6%), C14:0 (5.2%), C12:0 and summed feature 2 (3.7%; comprising C12:0 ALDE, iso-C16:1I and/or C14:0 3–OH and/or an unidentified fatty acid with equivalent chain length of 10.928), C12:03–OH(2.5%), C17:0 (1.6%), Iso-C17:0 (1.5%), Iso-C15:0 and C17:1ω8c (1.1%), and in minor amounts C13:0, C17:1ω6c, C18:0 and Unknown 12.484 (0.3–0.5%). The G + C content of the type strain (A-394T) is 48.2 mol%. The type strain is A-394T (= LMG 27910T = CAIM 1892T). It was isolated from the tissues of bleached Madracis decactis (Scleractinia) in St Peter & St Paul Archipelago, Brazil. Abbreviations

SPSPA St Peter & St Paul Archipelago

MLSA multilocus sequence analysis

AAI average amino acid identity

DDH DNA-DNA hybridization

GGDC Genome-To-Genome Distance Calculator

FAME fatty acid methyl ester analyses

MALDI-TOF matrix-assisted laser desorption/ionization time-of-flight

Supplemental Information

Table S1 GenBank accession numbers for genes and genomes

Upper GenBank accession numbers for the 16S rRNA gene, recA and rpoA housekeeping genes and genome sequences of Photobacterium sanctipauli sp. nov.; and for recA and rpoA of P. gaetbulicola LMG 27839T (data from this study). Lower Accession numbers for the Photobacterium strains’ genomes used for GGD calculation (data publicly available at GenBank).

Click here for additional data file.

Table S2 Strains and source information

Strains of Photobacterium sanctipauli sp. nov. and source information.

Click here for additional data file.

Table S3 Statistics and general features of the type strain’s genome

Statistics and general features of A-394T (= LMG 27910T = CAIM1892T) genome determined in RAST environment.

Click here for additional data file.

Table S4 Cellular fatty acids content of Photobacterium sanctipauli sp. nov. and related taxa of the genus Photobacterium

Taxa: 1, P. sanctipauli (range profile of five strains); 2, P. rosenbergii LMG 22223T (Thompson et al., 2005b); 3, P. gaetbulicola Gung 47T; and 4, P. lutimaris LMG 25278T (Kim et al., 2010). Summed feature 2 comprises C12:0 ALDE, iso-C16:1 I and/or C14:0 3-OH and/or an unidentified fatty acid with equivalent chain length of 10.928. Summed feature 3 comprises C16:1ω7c and/or iso-C15 2-OH. Data are expressed as percentages of total fatty acids. Fatty acids representing <1% are not shown, except for C18:0, because it was represented in all strains with approximately the same intensity.

Click here for additional data file.

Table S5 Phenotypic variability amongst strains of P. sanctipauli sp. nov

Phenotypic variability amongst representative strains of P. sanctipauli sp. nov. +, positive; −, negative; w, weak

Click here for additional data file.

Figure S1 recA phylogenetic tree

Neighbour-joining phylogenetic tree showing the position of P. sanctipauli sp. nov based on recA gene sequences (855 bp). The evolutionary distances were computed using the Kimura 2-parameter method. Phylogenetic analyses were conducted in MEGA5. Bootstrap values (>50%) shown are based on 1,000 repetitions. Vibrio maritimus R-40493T was used as outgroup. Bar, 2% estimated sequence divergence.

Click here for additional data file.

Figure S2 rpoA phylogenetic tree

Neighbour-joining phylogenetic tree showing the position of P. sanctipauli sp. nov based on rpoA gene sequences (969 bp). The evolutionary distances were computed using the Kimura 2-parameter method. Phylogenetic analyses were conducted in MEGA5. Bootstrap values (>50%) shown are based on 1,000 repetitions. Vibrio maritimus R-40493T was used as outgroup. Bar, 1% estimated sequence divergence.

Click here for additional data file.

Figure S3 MALDI-TOF MS fingerprint profiles of the 5 novel strains and the closely related type strains

Comparison of the MALDI-TOF MS fingerprint profiles of the 5 novel strains (A−394T, A-373, A-379, A-397 and A-398) showing they are not clonal. The closely related type strains of P. rosenbergii (LMG 22223T), P. gaetbulicola (LMG 27839T) and P. lutimaris (LMG 25278T) were included in the analysis. The dendrogram was constructed using Pearson correlation coefficient and UPGMA.

Click here for additional data file.

We are grateful to Leilei Li and Anneleen Wieme for their assistance with the MALDI-TOF MS data analysis.

Additional Information and Declarations

Competing Interests

Author Contributions

Field Study Permissions

DNA Deposition

Data Deposition

New Species Registration

Fabiano Thompson is an Academic Editor for PeerJ.

Ana Paula B. Moreira conceived and designed the experiments, performed the experiments, analyzed the data, wrote the paper, prepared figures and/or tables, reviewed drafts of the paper.

Gwen Duytschaever and Luciane A. Chimetto Tonon conceived and designed the experiments, performed the experiments, analyzed the data, prepared figures and/or tables, reviewed drafts of the paper.

Adriana M. Fróes analyzed the data, reviewed drafts of the paper.

Louisi S. de Oliveira performed the experiments, reviewed drafts of the paper.

Gilberto M. Amado-Filho contributed sampling, reagents/materials.

Ronaldo B. Francini-Filho conceived and designed the experiments, performed the experiments, contributed reagents/materials/analysis tools, reviewed drafts of the paper.

Paul De Vos and Jean Swings conceived and designed the experiments, contributed reagents/materials/analysis tools, reviewed drafts of the paper.

Cristiane C. Thompson contributed reagents/materials/analysis tools.

Fabiano L. Thompson conceived and designed the experiments, analyzed the data, contributed reagents/materials/analysis tools, reviewed drafts of the paper.

The following information was supplied relating to field study approvals (i.e., approving body and any reference numbers):

Sampling permit Sisbio no. 24732-1 issued by the Ministry of Environment Institute Chico Mendes (ICMBio).

The following information was supplied regarding the deposition of DNA sequences:

Nucleotide sequence data for Photobacterium sanctipauli sp. nov are available in the DDBJ/EMBL/GenBank databases under the following accession number(s): KC751065-6, KC751086, KC751088, KC751090-1 (16S rRNA); KF748538-41 (recA), KF748542-5 (rpoA) and ASHX00000000 for whole genome sequence. P. gaetbulicola LMG 27839T reported nucleotide sequence accession numbers: KF771650 (recA) and KF771651 (rpoA).

P.sanctipauli: This Whole Genome Shotgun project has been deposited at DDBJ/EMBL/GenBank under the accession JGVO00000000. The version described in this paper is version JGVO01000000.

The following information was supplied regarding the deposition of related data:

Gene sequences are deposited at http://www.taxvibrio.lncc.br/, Genbank.

The following information was supplied regarding the registration of a newly described species:

The type strain A-394T have been deposited in two culture collections:

BCCM/LMG Bacteria Collection (Belgium) = LMG 27910T,

and

Collection of Aquatic Important Microorganism CAIM (Mexico) = CAIM 1892T.

Cultures are also deposited in the brazilian collection Coleção de Bactérias do Ambiente e Saúde http://cbas.fiocruz.br/ for open access.

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
