# Peer review of "Photobacterium sanctipauli sp. nov. isolated from bleached Madracis decactis (Scleractinia) in the St Peter & St Paul Archipelago, Mid-Atlantic Ridge, Brazil"

_PeerJ, doi:10.7717/peerj.427_

## Round 0.1 · original submission · Minor Revisions

The paper is well written and very clearly presented. I think the reviewers' comments are straightforward, but please make sure you address them as directly as possible.

·

Basic reporting

This brief paper describes the type strain definition of a new species of Photobacterium, which the authors have called Photobacterium sanctipauli after the location where it was found. This is a short paper that thoroughly describes the strain, is a very standard paper typical of one that would be in a journal of systematics, and I think it is and is appropriate for publication in PeerJ.

Experimental design

The experimental design uses the typical approaches for defining microbial taxonomy of new species of bacteria that are isolated. I do not see any shortcomings with the experimental design or areas that they have missed.

Validity of the findings

The findings are appropriate for the description of a newly identified microbial species.

Additional comments

Abstract
Line 1. Start with telling me the organism. Five novel strains of Photobacterium
Line 3-4. but no strains were related to them

Introduction

line 18- should read: Coral microbiologists are challenged to increase our understanding in order to mitigate the worldwide spread of infectious diseases that are implicated in the decrease of coral cover in reef systems, markedly associated with climate changes and anthropogenic driven environmental disturbances

·

Basic reporting

No comments

Experimental design

No comments

Validity of the findings

No comments

Additional comments

Generally well-written manuscript, sound data, conclusions justified.

Minor suggestions for improvement include

1. Improve the English at selected locations :
-abstract : none related > no related
-line 3 : source isolation > isolation source
-lines 23-24 : rephrase, sounds awkward

2. Lines 91-92. Do the authors mean that the other close phylogenetic neighbours have not been validly described yet? Rephrase sentence.

3. Line 105 and further. Provide additional information on how in silico DDH values were calculated. Is this based on ANI ? Provide appropriate reference(s).

4. Lines 114-116. The fact that the 5 strains are different is not a guarantee that they are a good representation of the overall phenotype. Rephrase.

---

## Round 0.2 · accepted · Accept

I am very sorry it has taken longer than normal from submission of the revised paper to the editorial decision; however, I have been out of email contact for an extended period, during which one of the reviewers kindly checked the revised MS. There are still one or two minor errors/typos (spelling), which I trust will be corrected in the further processing of the MS for pubication.